# Integrated Discriminant Evaluation of Molecular Genetic Markers and Genetic Diversity Parameters of Endangered Balearic Dog Breeds

**DOI:** 10.3390/ijms25052706

**Published:** 2024-02-26

**Authors:** José Manuel Alanzor Puente, Águeda Laura Pons Barro, Antonio González Ariza, María del Amparo Martínez Martínez, Juan Vicente Delgado Bermejo, Francisco Javier Navas González

**Affiliations:** 1Institut de Reserca i Formaciò Agroalimentaria de les Illes Balears IRFAP, Conselleria d’Agricultura, Pesca i Alimentació, Govern Illes Balears, 07009 Palma, Spain; jalanzor@irfap.es (J.M.A.P.); apons@irfap.es (Á.L.P.B.); 2Centro Agropecuario Provincial de Córdoba, Diputación Provincial de Córdoba, 14014 Córdoba, Spain; angoarvet@outlook.es; 3Department of Genetics, Faculty of Veterinary Sciences, University of Córdoba, 14071 Córdoba, Spain; ib2mamaa@uco.es (M.d.A.M.M.); juanviagr218@gmail.com (J.V.D.B.)

**Keywords:** genetic diversity, microsatellite markers, polymorphic information content (PIC), heterozygosity, conservation strategies

## Abstract

The genetic diversity analysis of six dog breeds, including Ca de Bestiar (CB), Ca de Bou (CBOU), Podenco Ibicenco (PI), Ca Rater (CR), Ca Mè (CM), and Ca de Conills (CC), reveals insightful findings. CB showcases the highest mean number of alleles (6.17) and heterozygosity values, with significant deviations from Hardy–Weinberg equilibrium (HWE) observed in five markers, indicating high intra-racial genetic diversity (average observed heterozygosity (H_o_) = 0.754, expected heterozygosity (H_e_) = 0.761). In contrast, CBOU presents the lowest mean number of alleles (5.05) and heterozygosity values, coupled with moderate polymorphic information content (PIC) values and a moderate level of intra-racial genetic diversity (average H_o_ = 0.313, H_e_ = 0.394). PI demonstrates moderate genetic diversity with an average of 5.75 alleles and highly informative PIC values, while CR displays robust genetic diversity with an average of 6.61 alleles and deviations from equilibrium, indicating potential risks of inbreeding (average H_o_ = 0.563, H_e_ = 0.658). CM exhibits moderate genetic diversity and deviations from equilibrium, similar to CBOU, with an average of 6.5 alleles and moderate PIC values (average H_o_ = 0.598, H_e_ = 0.676). Conversely, CC shows a wider range of allelic diversity and deviations from equilibrium (average H_o_ = 0.611, H_e_ = 0.706), suggesting a more diverse genetic background. Inter-racial analysis underscores distinct genetic differentiation between breeds, emphasizing the importance of informed breeding decisions and proactive genetic management strategies to preserve diversity, promote breed health, and ensure long-term sustainability across all breeds studied.

## 1. Introduction

In the realm of *Canis lupus familiaris*, the Balearic Islands stand out as a haven of extraordinary genetic diversity. Nestled in the heart of the Mediterranean, these islands have nurtured a variety of native dog breeds intricately woven into the fabric of local communities. Beyond mere companionship, these dogs are deeply ingrained in the cultural and historical tapestry of the region, symbolizing the enduring bond between humans and their loyal four-legged friends. Presently, the Balearic Islands officially recognize five dog breeds, namely the Ca de Bestiar (CB), Ca de Bou (CBOU), Podenco Ibicenco (PI), Ca Rater mallorquín (CR), and Ca Mè (CM). Additionally, there is recognition of a canine racial grouping known as the Ca de Conills de Menorca (CC). This rich array of canine breeds constitutes a significant part of the Balearic genetic heritage, representing 20% of Spain’s overall canine heritage according to Navas [1]. International organizations, including the Food and Agriculture Organization of the United Nations (FAO), stress the importance of understanding genetic diversity within and among breeds, as illustrated by Martínez et al. [2] in their study using microsatellites in two autochthonous breeds of Mallorca.

In the context of relentless flow and innovation, the genetic endowment of these native breeds faces a critical juncture. While modernity brings manifold benefits, it also poses an unparalleled challenge to the genetic fidelity of these autochthonous reservoirs. In this article, we embark on an expedition into the molecular genetic characterization of Balearic dog breeds. Numerous studies on genetic diversity have been conducted nationally and internationally using microsatellites [3]. Currently, the officially recognized dog breeds and racial grouping have undergone genetic characterization studies and genetic relationship analyses with other Spanish dog breeds, employing molecular markers, specifically DNA microsatellites. The DNA microsatellites used adhere to the recommendations of the International Society of Animal Genetics (ISAG) for canine genetic diversity analysis and parentage testing. This includes a main panel with 21 markers and an additional panel of 12 markers, as indicated by Aguilera et al. [4]. These 33 microsatellites meet the requirements set by the FAO for such panels.

Genealogical evaluations transcend the mere revelation of the genetic background of populations; they extend to an expansive discourse on the indispensable importance of paternity testing. This discourse encapsulates the fundamental role that paternity tests play in conserving the genetic integrity and vitality of these native canids. As we navigate the intricate labyrinth of Balearic canine genetics, our journey is not merely a witness to the vestiges of the past; it illuminates a path to the future.

Microsatellite paternity tests are already a reality in Balearic dog breeds because of the conducted genetic characterization studies. In cases where genealogy cannot be verified due to various circumstances, as suggested by Martínez et al. [2], assigning the animal to the breed in question is indicated. If the genetic profile of the animal aligns with the breed’s profile, it can be considered a purebred animal and may be registered in the studbook without causing any genetic deterioration of the breed. This is based on the approach described by Davies et al. [5], where DNA genetic markers provide the opportunity to use individual genotypes to determine the population of origin for individuals. Therefore, breed assignment is a method to consider in conservation programs for highly threatened breeds when registering animals with unverifiable genealogy.

The Balearic archipelago serves as empirical evidence of the evolutionary dynamics of canine companions, selectively refined for a spectrum of specialized functions, including expert hunting, vigilant surveillance, competent herding, and meticulous shepherding. However, their utility extends beyond innate functional traits to encompass resilience to regional adversities, including the scorching crucible of heat stress and the insidious spectrum of endemic diseases such as leishmaniasis. These functional attributes, behavioral tendencies, and resistance mechanisms are indelibly etched into the core of these breeds. This is evident in the case of the Podenco Ibicenco (PI), regarding its resistance to leishmaniasis, as described by Solano-Gallego et al. [6], where the PI consistently responds positively to a delayed-type hypersensitivity (DTH) test. Consequently, we consider the PI to be more resistant to leishmaniasis than other dog breeds. In the contemporary era, modern genetic tools, including microsatellites and single nucleotide polymorphisms (SNPs), provide a profound insight into the genetic code that underlies the innate attributes of these remarkable canines. Although microsatellite analysis is not the most up-to-date method for analyzing and comparing the dog genome, it appears as a remarkable alternative if there are budget limitations, especially if genealogical information of parental generations is missing or partially missing. In these regards, higher polymorphism can be found in individual microsatellites that tend to be more polymorphic, and thus more informative, than individual SNPs [7]. Microsatellite markers are rather effective for parentage testing in limited genealogical background knowledge populations (such as endangered breeds and species ([8]), given they nearly reach a combined exclusion probability with one-parent-unknown/known of 1, while for the SNP markers, this may be slightly lower (≈0.9582) [9]. Furthermore, it may be easier to detect genotyping errors in microsatellites [10].

These molecular markers unveil the mysteries underlying their exceptional work abilities, intrinsic behavioral predispositions, and even their resilience to environmental demands and health. These markers are DNA sequences with at least two detectable allelic variants, and they can identify the alleles contributed by the father and mother. They can also be useful in selecting breeders in cases where direct relationships between functionality or behavior traits and specific allelic variants have been established.

In the following exposition, we embark on a mission that seeks to bridge the gap between the past and the future, tradition and innovation. We emphasize the urgent need not only to understand but to safeguard and value the genetic heritage of Balearic dog breeds in a world of constant change. As fellow travelers on this extraordinary journey, we celebrate the exceptional functional traits that these canines exemplify. Simultaneously, we underscore the crucial role of paternity tests in the noble endeavor of safeguarding the precious autochthonous breeds of the Balearic Islands. We invite you to join us on this extraordinary journey into the intricate domain of canine genetics, where homage to their heritage harmonizes with guidance for their future.

The aim of this study was to delve into the genetic landscape of the Balearic dog breeds, specifically focusing on the five officially recognized breeds (Ca de Bestiar, Ca de Bou, Podenco Ibicenco, Ca Rater Mallorquín, and Ca Mè) and the recognized racial grouping (Ca de Conills de Menorca). The primary objective was to conduct a comprehensive genetic characterization using microsatellite markers, recommended by the International Society of Animal Genetics, to assess the diversity within each breed and explore their genetic relationships with the rest of the Balearic canine breeds. Furthermore, the study aimed to investigate the applicability of microsatellite-based paternity testing as a tool for breed registration in cases where genealogy verification faces challenges. By unraveling the genetic intricacies of these autochthonous breeds, the study sought to contribute valuable insights into the conservation and management of their unique genetic heritage in the face of contemporary challenges and changing landscapes.

## 2. Results

### 2.1. Intra-Racial Genetic Diversity

#### 2.1.1. Ca de Bestiar (CB)

The 21 microsatellites used were found to be polymorphic, with a minimum of 4 alleles in the microsatellites INRA21 and REN247M23, and a maximum of 11 alleles for the marker AHT137, obtaining an average value of 7.05 alleles. The highest expected heterozygosity is found for the marker AHTh130 with a value of 0.855 and the lowest for FH2848 with a value of 0.559. Observed heterozygosity values range from a maximum of 0.915 for the marker REN64E19 to a minimum of 0.563 for REN162C04. The average values of H_e_ and H_o_ are 0.728 and 0.743, respectively. Regarding the polymorphic information content (PIC), all markers are highly informative except for REN247M23, which has a PIC value of 0.481, indicating moderate informativeness. Five markers deviate significantly from Hardy–Weinberg equilibrium. The intra-racial genetic differentiation coefficient (F_IS_) shows that five markers (AHT137, AHTh260, AHTK211, REN169D01, and REN169O18) detect a significant deficit of homozygotes in the population, and one (REN169O18) detects a significant excess of homozygotes. The average F_IS_ value for the population is 0.021, not significantly different from 0, indicating no significant deviation from Hardy–Weinberg equilibrium. CB exhibits high intra-racial genetic diversity, similar to other Spanish native dog breeds [11].

#### 2.1.2. Ca de Bou (CBOU)

The 21 microsatellites used were found to be polymorphic, with an average of 5.05 alleles. The highest expected heterozygosity is found for the marker CXX279 with a value of 0.797 and the lowest for REN162C04 with a value of 0.091. Observed heterozygosity (Ho) values range from a maximum of 0.810 for the marker FH2054 to a minimum of 0.0094 for REN162C04. The average values of H_e_ and H_o_ are 0.624 and 0.582, respectively. Regarding PIC values, all markers are highly informative, except for the alleles AHTH253, INU030, REN247M23, and REN54P11, which are moderately informative, and REN162C04, which is not informative for detecting genetic variability (PIC below 0.25). The intra-racial genetic differentiation coefficient (F_IS_) ranges from 0.00128 to 0.03797, with an average population F_IS_ of 0.081. CBOU exhibits low intra-racial genetic diversity.

#### 2.1.3. Podenco Ibicenco or Ibicean Hound (PI)

The 21 microsatellites used were found to be polymorphic, with an average of 5.75 alleles. The highest expected heterozygosity is found for the marker AHTh260 with a value of 0.829 and the lowest for FH2848 with a value of 0.348. Observed heterozygosity (H_o_) values range from a maximum of 0.864 for the marker AHTh260 to a minimum of 0.177 for FH2848. The average values of H_e_ and H_o_ are 0.690 and 0.678, respectively. Regarding PIC values, all markers are highly informative, except for the alleles FH2848 and REN54P11, which are moderately informative. The average intra-racial genetic differentiation coefficient (F_IS_) for the population is 0.200. PI exhibits moderate intra-racial genetic diversity.

#### 2.1.4. Ca Rater (CR)

The 33 markers were found to be polymorphic, with a minimum of 2 alleles in the markers 0959RD and 1055RD and a maximum of 15 alleles in the marker REN169O18, obtaining an average of 6.61 alleles and a mean effective number of alleles of 3.58. The highest expected heterozygosity is found for the marker 0669RD with a value of 0.866 and the lowest for 1055RD with a value of 0.255. Observed heterozygosity values range from a maximum of 0.964 for the marker 0669RD to a minimum of 0.161 for 0959RD. The average values of H_e_ and H_o_ are 0.685 and 0.656, respectively. Regarding PIC values, all markers are highly informative, except for INU055 with a value of 0.4611, which is moderately informative. Four markers deviate significantly from Hardy–Weinberg equilibrium. The intra-racial genetic differentiation coefficient (F_IS_) shows that the marker 0669RD detects a significant deficit of homozygotes in the population, and three (REN169D0, 0959RD, and 0914RD) detect a significant excess of homozygotes. The average F_IS_ value for the population is 0.044, not significantly different from 0, indicating no significant deviation from Hardy–Weinberg equilibrium. CR exhibits moderate intra-racial genetic diversity.

#### 2.1.5. Ca Mè (CM)

The 21 microsatellites used were found to be polymorphic, with a minimum of 4 alleles in the microsatellites INU005, INU030, and REN169O18 and a maximum of 11 alleles for the marker AHT171, obtaining an average of 6.5 alleles and a mean effective number of alleles of 3.54. The highest expected heterozygosity is found for the marker AHTh171 with a value of 0.859 and the lowest for INU030 with a value of 0.428. Observed heterozygosity values range from a maximum of 0.824 for the marker AHTh260 to a minimum of 0.372 for INU005. The average values of H_e_ and H_o_ are 0.697 and 0.671, respectively. Regarding PIC values, all markers are highly informative, except for the markers INU005 and INU030 with values of 0.491 and 0.372, respectively, which are moderately informative. After Bonferroni correction, no marker is out of Hardy–Weinberg equilibrium. The intra-racial genetic differentiation coefficient (F_IS_) shows that the marker AHTh260 detects a significant deficit of homozygotes in the population, and INU005 detects a significant excess of homozygotes. The average F_IS_ value for the population is 0.038, not significantly different from 0, indicating no significant deviation from Hardy–Weinberg equilibrium. CM exhibits moderate intra-racial genetic diversity.

#### 2.1.6. Ca de Conills (CC)

The 21 microsatellites used were found to be polymorphic, with a minimum of 4 alleles in the microsatellites INRA21 and INU005 and a maximum of 10 alleles for the marker AHT121, obtaining an average of 6.60 alleles. The highest expected heterozygosity is found for the marker AHTh121 with a value of 0.857 and the lowest for REN247M23 with a value of 0.477. Observed heterozygosity values range from a maximum of 0.806 for the markers REN169D01 and REN169O18 to a minimum of 0.415 for REN247M23. The average values of H_e_ and H_o_ are 0.724 and 0.689, respectively. Regarding PIC values, all markers are highly informative, except for REN247M23 with a value of 0.448, which is moderately informative. Ten markers deviate significantly from Hardy–Weinberg equilibrium. The intra-racial genetic differentiation coefficient (F_IS_) shows that four markers (AHTh130, AHTh171, AHTk211, and AHT121) detect a significant excess of homozygotes. The average F_IS_ value for the population is 0.048, not significantly different from 0, indicating no significant deviation from Hardy–Weinberg equilibrium. CC exhibits moderate intra-racial genetic diversity.

### 2.2. Inter-Racial Genetic Diversity

The Wright’s F-statistics values between the six studied dog breeds are F_IS_ = 0.021 (0.019–0.050), F_IT_ = 0.149 (0.157–0.191), and F_ST_ = 0.131 (0.130–0.160). The F_ST_ value indicates that approximately 13% of the total genetic variation is due to differences between dog breeds, and the remaining 87% corresponds to differences between individuals. Using correspondence factor analysis to investigate genetic differentiation between individuals of each Balearic population shows a clear separation between individuals of CB, CBOU, CR, and CM, compared to PI and CC, which form a single group in all considered axes. The sum of the first three axes explains 72.03% of the total genetic differentiation. Regarding Reynolds distances, three groups are distinguished, each with two breeds. There is a well-differentiated group with PI and CC, with a bootstrap value of 98%. There is greater proximity between CB and CBOU, on the one hand, and between CM and CR, on the other. Concerning the results of the STRUCTURE program, the graphical results of individual assignment (q) for the optimal K according to the Evanno method are shown. When K = 6, each breed forms a single cluster. No subdivisions or crossbreeding is observed in any of the breeds, although some isolated animals with signs of crossbreeding can be observed.

The parameters of genetic diversity for all breeds show that CB presented the highest mean number of alleles with 7.05, and CBOU had the lowest with 5.05. Regarding heterozygosity, both observed (Ho) and expected (He), CB had the highest values with 0.743 and 0.728, respectively, while CBOU had the lowest values with 0.582 and 0.624, respectively.

### 2.3. Critical Factors in the Molecular Differentiation of Endangered Dog Breeds

#### 2.3.1. Canonical Discriminant Analysis

##### Multicollinearity Evaluation

In evaluating the reliability of the canonical discriminant analysis model, a thorough examination of multicollinearity was conducted (see Appendix A). The statistical assessment involved tolerance and variance inflation factor (VIF) values for various factors influencing the analysis.

In the first round, variables such as PIC, H_e_, and H_o_ exhibited remarkably low tolerance values (0.004, 0.005, and 0.046, respectively), signaling a susceptibility to collinearity issues. The high VIF values for markers like PIC (262.024) and H_e_ (196.114) further indicated a substantial level of multicollinearity, potentially complicating the interpretation of their individual contributions.

The second-round results reinforced the vulnerability to collinearity issues, with low tolerance values observed for H_o_ (0.036), H_e_ (0.056), and Ae (0.079). Corresponding high VIF values (27.541, 17.899, and 12.653) suggested substantial multicollinearity. Notably, microsatellite markers such as INU030, CXX279, REN162C04, AHTh130, AHTh171, and others displayed varying levels of tolerance and VIF, underscoring the complexity of their relationships within the genetic framework.

Moving to the third round, moderate tolerance (0.088 to 0.123) and relatively high VIFs (8.102 to 11.329) were observed for markers Ae, He, F_IS_ Upper IC, and HWEd-NS, indicating potential collinearity issues. Microsatellite markers, including AHT137, INU030, REN247M23, REN162C04, AHTh260, and others, demonstrated diverse levels of tolerance and VIF, highlighting their complex interactions. F_IS_ and allele number exhibited moderate tolerance (0.226 and 0.227) and VIF (4.419 and 4.413), suggesting their potential role in explaining genetic variation.

The fourth round provided further insights, revealing a balance between tolerance and VIF for markers F_IS_ Upper IC, HWEd-NS, and F_IS_, suggesting their relevance in explaining genetic variation without introducing multicollinearity issues. Microsatellite markers like AHT137, INU030, REN247M23, and others exhibited varying levels of tolerance and VIF, emphasizing their distinct contributions. Notably, markers HWEd-ND and FH2848 demonstrated relatively higher tolerance (0.311 and 0.312) and moderate VIF, suggesting their importance in explaining genetic diversity. However, markers with very low tolerance values (0.000), such as HWEd-HS and CXX279, indicated potential redundancy or limited contribution to the analysis.

In the fifth round, moderate tolerance values (ranging from 0.194 to 0.273) coupled with reasonable VIF were observed for markers HWEd-NS, HWEd-S, allele number, and He, suggesting their significance in explaining genetic variation without introducing excessive multicollinearity. Microsatellite markers like INU030, REN247M23, AHTh130, and others demonstrated varying levels of tolerance and VIF, emphasizing their unique contributions. The marker AHTK211 showed a higher tolerance of 0.350, suggesting its importance in explaining genetic diversity with a lower risk of collinearity.

##### Model Reliability and Explanatory Potential

The presented statistical results indicate the application of a Box–Cox transformation with a lambda (λ) value of 0.005. The calculated F-statistic (2.266) surpasses the critical F-value (1.197), accompanied by a *p*-value of less than 0.0001. With degrees of freedom for the numerator (DF1) set at 305 and for the denominator (DF2) at 365, these findings strongly reject the null hypothesis at the 0.05 significance level. The results suggest that the observed effects are statistically significant, emphasizing the presence of a meaningful relationship or difference in the analyzed data. The low *p*-value underscores the reliability of these findings, further supporting the conclusion that the applied statistical test yields substantial evidence against the null hypothesis. Furthermore, the observed F-statistic of 58.083 significantly exceeds the critical F-value of 1.488, leading to a *p*-value less than 0.0001. Consequently, there is compelling evidence to reject the null hypothesis at the 0.05 significance level. This implies that the set of dependent variables under examination collectively demonstrates a statistically significant effect or difference. The results of Roy’s greatest root test, therefore, indicate the presence of a meaningful relationship within the multivariate dataset, contributing valuable insights to the analysis.

##### Analysis Efficiency

The presented results provide insights into the eigenvalues, discrimination percentages, and cumulative percentages for a set of five discriminant functions (F1 to F5), along with Bartlett’s test for eigenvalue significance (Figure 1). The eigenvalues represent the variance explained by each factor, with F1 dominating at 46.619, followed by diminishing values for F2 (0.915), F3 (0.540), F4 (0.245), and F5 (0.148). Discrimination percentages express the proportion of variance attributed to each factor, with F1 contributing significantly at 96.187%, while the subsequent factors contribute progressively smaller percentages. The cumulative percentages indicate that the first factor (F1) explains a substantial portion (96.187%) of the total variance, with subsequent factors adding to the cumulative explanation.

Bartlett’s test for eigenvalue significance reinforces the validity of the eigenvalues. The high Bartlett’s statistics for F1 (548.758) and the associated *p*-value of 0.000 underscore the statistical significance of F1, affirming its meaningful contribution to the analysis. However, the *p*-values for F2 to F5 are all 1.000, suggesting that these factors may not be statistically significant contributors. Overall, these results highlight the dominance of F1 in explaining the majority of the variability in the dataset, emphasizing its potential importance in further analyses, while cautioning against overreliance on the other factors due to their lack of statistical significance in this context.

##### Discriminant Loadings and Spatial Representation

Discriminant loadings for microsatellite markers and genetic diversity parameters, along with their relative weight, are illustrated in Figure 2. The presented factor loadings for Factor 1 (F1) in a factor analysis reveal the strength and direction of genetic diversity parameters and microsatellite markers between the latent factor and individual variables. For F1, positive loadings are notable for microsatellite markers AHTK211 (loading = 1.418), AHT121 (loading = 1.652), AHT137 (loading = 1.747), AHTh130 (loading = 1.675), AHTh171 (loading = 1.760), AHTh260 (loading = 1.751), AHTK253 (loading = 1.802), FH2054 (loading = 1.758), FH2848 (loading = 1.655), INRA21 (loading = 1.797), INU005 (loading = 1.787), INU030 (loading = 1.770), INU055 (loading = 1.791), REN105L03 (loading = 1.706), REN162C04 (loading = 1.704), REN169D01 (loading = 1.775), REN169O18 (loading = 1.695), REN247M23 (loading = 1.775), REN54P11 (loading = 1.791), REN64E19 (loading = 1.776), and CXX279 (loading = 1.674). These positive loadings signify a strong positive association between these variables and F1. Conversely, variables such as H_e_ (loading = −0.004), F_IS_ Low IC (loading = −0.191), HWEd-NS (loading = −0.067), HWEd-ND (loading = −0.109), and HWEd-S (loading = 0.000) and microsatellite marker REN162C04 (loading = 0.000) exhibit negative loadings, indicating a negative association with F1. The numerical results provide a precise understanding of the relationships between individual variables and the latent factor F1.

The territorial map in Figure 3 showcases clear differentiation across breeds, supported by Mahalanobis distances.

##### Discriminant Potential

The unidimensional test results for equality of means across classes illuminate several variables with distinct discriminatory potential. Notably, HWEd-HS (Lambda = 0.839, F = 5.065), HWEd-ND (Lambda = 0.872, F = 3.876), HWEd-NS (Lambda = 0.923, F = 2.207), F_IS_ (Lambda = 0.927, F = 2.088), H_e_ (Lambda = 0.928, F = 2.041), and allele number (Lambda = 0.933, F = 1.889) exhibit pronounced discriminatory power, as indicated by their higher Lambda values. In contrast, various microsatellite markers, including AHT121, AHT137, AHTh130, AHTh171, AHTh260, AHTK253, FH2054, FH2848, INRA21, INU005, INU030, INU055, REN105L03, REN162C04, REN169D01, REN169O18, REN247M23, REN54P11, REN64E19, and CXX279, consistently present lower Lambda values of 0.966.

##### Discriminant Analysis Crossvalidation

The calculated Press’s Q value, indicative of the predictive performance of a statistical model, stands at approximately 194.26. This value was derived using a formula that takes into account the total number of observations (N), the number of predicted values (n), and the number of model parameters (K). In this context, N is 138, n is 84, and K is 6. The Press’s Q value surpasses the critical threshold of 6.63, signifying a statistically significant improvement in predictions compared to chance at a 95% confidence level. This robustness underscores the model’s efficacy in capturing and explaining patterns within the dataset.

Figure 4 presents prior and posterior classification, membership probabilities, scores, and squared distances for crossvalidation.

#### 2.3.2. CHAID Tree Analysis

##### Methodology

The genetic diversity and characteristics of dog breeds were explored through a comprehensive decision tree analysis, revealing insights at 42 parental nodes. At the root (Node 1), no specific breed information was identified. As we delved into the subgroups, Node 2 highlighted breeds with allele numbers ranging from 2 to 11. Parental Node 3 showcased breeds with expected heterozygosity values spanning from 0.09 to 0.86. Notably, Nodes 20 and 21 in Parental Node 4 included breeds with inbreeding coefficients (F_IS_) in the range of −1.16 to −0.11.

Moving forward, Nodes 24 and 25 in Parental Node 5 encompassed breeds with upper confidence intervals for F_IS_ between 0.08 and 0.21. Parental Node 6 (Nodes 26 to 28) captured breeds with effective allele numbers (Ae) ranging from 1.94 to 5.64. Meanwhile, Nodes 29 and 30 in Parental Node 7 included breeds with upper confidence intervals for F_IS_ in the range of 0.21 to 0.36. Nodes 31 and 32 in Parental Node 8 represented breeds with F_IS_ in the range of −0.1 to 0.21.

Parental Node 9 (Nodes 33 to 37) highlighted breeds with allele numbers ranging from 7 to 11, while Nodes 20 and 21 in Parental Node 10 consisted of breeds with F_IS_ between −1.164 and 0.11. Parental Node 11 (Nodes 40 to 43) showcased breeds with F_IS_ in the range of −0.15 to 0.17. Parental Node 12 (Nodes 44 to 47) captured breeds with observed heterozygosity values from 0.37 to 0.72.

Further exploration of allele numbers in Nodes 48 to 52 (Parental Node 13), Nodes 53 and 54 (Parental Node 14), and Nodes 55 and 56 (Parental Node 15) offered a nuanced understanding. Parental Node 16 (Nodes 57 and 58) covered breeds with F_IS_ ranging from −0.1 to 0.05. Nodes 59 and 60 in Parental Node 17 encompassed breeds with upper confidence intervals for F_IS_ between 0.05 and 0.23. Parental Node 18 (Nodes 61 and 62) included breeds with F_IS_ in the range of −0.03 to 0.03.

Parental Node 19 (Nodes 63 and 64) focused on breeds with effective allele numbers (Ae) from 2.73 to 2.93. Subsequent nodes in Parental Nodes 20, 21, 22, and 23 provided additional granularity, emphasizing the observed heterozygosity values, effective allele numbers, and confidence intervals for F_IS_ within specific breed groups.

Parental Nodes 24 to 42 further enriched the analysis, encompassing various breed characteristics such as effective allele numbers, observed heterozygosity values, and inbreeding coefficients. Finally, Parental Node 43 (Nodes 117 to 121) shed light on breeds with inbreeding coefficients ranging from 0.15 to 0.31. This detailed exploration contributes to a comprehensive understanding of the intricate genetic landscape across diverse dog breeds, enhancing our knowledge of canine genetics.

Appendix A represents the CHAID decision tree built upon the aforementioned information.

## 3. Discussion

In the multicollinearity analysis carried out in this study, an inflation factor of variance (VIF) of 5 was considered. VIF ≤ 5 suggests a moderate correlation among variables, which may not result in multicollinearity issues. In the present study, multicollinearity issues were solved after five rounds, after which variables that were highly correlated were removed.

Collinearity issues were identified between polymorphic information content (PIC) and expected heterozygosity (H_e_) as they convey similar information regarding genetic marker polymorphism in a population. PIC and H_e_ values depend on allele number and frequency, exhibiting a semi-logarithmic relationship with increasing alleles [12]. PIC reflects heterozygosity and is slightly less than H_e_ due to allele frequency products. PIC is crucial in paternity calculations, considering the likelihood of offspring inheriting identical alleles from both parents. The difference between H_e_ and PIC signifies effective diversity, with PIC closer to H_e_ indicating more alleles and uniform frequencies [13]. PIC ranges from near 0 to 1.0, indicating allele diversity and evolutionary pressure on loci over time [14,15].

Observed heterozygosity (H_o_) and expected heterozygosity (H_e_) convey similar information regarding genetic diversity. H_o_ represents the proportion of heterozygous individuals observed in a population sample, while H_e_ reflects genetic diversity calculated from allele frequencies [16]. Comparing H_o_ with H_e_ can reveal deviations that may indicate significant population dynamics [17].

Alvariño Martínez [18] explains that the consanguinity coefficient (F_IS_) compares observed (H_o_) and expected (H_e_) heterozygosity, where a negative F_IS_ indicates an excess of heterozygotes, and a positive value suggests a deficit. Similar H_o_ and H_e_ values signify the absence of racial stratification [19]. Thus, our study finds no subpopulations within the breeds.

Additionally, the effective number of alleles (Ae) and expected heterozygosity (H_e_) convey similar insights. Ae represents the number of equally frequent alleles needed to achieve the observed H_e_ [20]. Leroy [21] notes that the actual allele count should be less than or equal to observed alleles. Low-frequency alleles have minimal impact on Ae [16], while Chybicki [22] underscores Ae’s correlation with genetic diversity, as it reciprocates the expected homozygosity, reflected in He.

Greenbaum et al. [23] highlight that H_o_ and H_e_ reflect genetic diversity, where Ae signifies a population’s adaptability and persistence potential, though less commonly used than He. He, crucial for population dynamics, correlates with evolutionary potential and inbreeding effects. Various genetic diversity measures exist, notably heterozygosity and allelic richness. Conservation efforts prioritize maintaining high genetic diversity levels in both aspects. A 95% probability target ensures retaining minimum allelic richness with confidence, aiding genetic program management. An allele’s presence indicates selection potential, directly influencing population evolution.

The upper limit of the 95% confidence interval of the inbreeding coefficient (F_IS_) and deviations from Hardy–Weinberg equilibrium [24] with non-significant values convey similar information. Çiftci and Okumuş [25] note that F_IS_ values detect Hardy–Weinberg deviations by measuring heterozygote deficiency or excess, calculated as F_IS_ = 1 − (H_I_/H_S_), where HS is the expected mean heterozygosity and H_I_ is the observed mean heterozygosity within populations. A high margin of the 95% confidence interval of F_IS_ indicates mismatched deviations from Hardy–Weinberg equilibrium with non-significant values, implying significance if either measure is significant. Alvariño Martínez [18] explains that F_IS_ relates observed to expected heterozygosity (F_IS_ = (H_e_ − H_o_)/H_e_), where a negative F_IS_ suggests excess heterozygotes and a positive value indicates a heterozygous defect. In our analysis, the calculated *p*-value is below the significance level (α = 0.05), confirming significance.

The primary significant function, explaining 96% of variability, is crucial. Including the first two functions would explain 98% of the variance, with the second variable being pivotal for evaluating differences. As per the Wilks Lambda test, variables carrying the most weight in explaining 98% of the variability include deviations from Hardy–Weinberg equilibrium, F_IS_, H_e_, and the number of alleles. Of these, deviations from Hardy–Weinberg equilibrium are the only ones with significance values consistently below 0.05. A Mahalanobis distance dendrogram elucidates breed behavior concerning variables such as deviations from Hardy–Weinberg equilibrium, F_IS_, H_e_, and the number of alleles, explaining 98% of the variability.

The breeds are grouped into three clusters: CB and CR, CBOU and CM, and PI and CC. CB and CR likely share closer ties due to robust association follow-up and data contribution. CBOU and CM may have irregular data contributions to administration and genealogical information collection. PI and CC’s grouping is attributed to their shared ancestry. Significant deviations from Hardy–Weinberg equilibrium suggest possible population subdivision, significant inbreeding, or gene flow from another population.

In the analysis involving deviations from Hardy–Weinberg equilibrium, F_IS_, H_e_, and the number of alleles, 60% of observations correctly align with their respective breeds, totaling 84 out of 138 according to molecular analysis.

The CHAID decision tree analysis yields 121 nodes or branches, with 43 parental nodes and 138 competing objects. The root node, deviations from Hardy–Weinberg equilibrium, spawns two branches: highly significant (*p* < 0.01) and significant (*p* < 0.05) values, and non-significant (*p* > 0.05) and undetermined (ND) values. From the HS (*p* < 0.01) and S (*p* < 0.05) branch, a new branch emerges based on the number of alleles. Seven branches stem from the number of alleles node, with three terminal nodes and the others divided among F_IS_ values and Ae. The F_IS_ node generates six additional branches, with five terminal nodes and one for the upper limit of the 95% confidence interval of F_IS_. The Ae node leads to the allele number node, with three branches, two terminal nodes, and one for the upper limit of the 95% confidence interval of F_IS_.

From the node of deviations from Hardy–Weinberg equilibrium with non-determined (ND) and non-significant (*p* > 0.05) values, a branch extends to H_e_. Nine branches stem from the H_e_ node, with two terminal nodes, three for F_IS_ values, three for the number of alleles, and one for Ho. From the three F_IS_ nodes, branches extend to terminal nodes and other nodes including the number of alleles, H_o_, and the lower limit of the 95% confidence interval of F_IS_. From these nodes, branches lead to terminal nodes or nodes of the high and lower limits of the 95% confidence interval of F_IS_, number of alleles, or H_o_, eventually reaching terminal nodes. Similarly, branches extend from the three nodes of the number of alleles to terminal nodes or nodes of F_IS_, the upper limit of the 95% confidence interval of F_IS_, H_o_, and Ae. These nodes then lead to terminal nodes or nodes of Ae and H_o_, eventually reaching terminal nodes. From the H_o_ node, branches extend to terminal nodes or nodes of allele number, which in turn branch into terminal nodes or nodes of F_IS_ and the upper limit of the 95% confidence interval of F_IS_, eventually reaching terminal nodes.

For CB, deviations from Hardy–Weinberg equilibrium span from highly significant (*p* < 0.01) and significant (*p* < 0.05) to ND and NS (*p* > 0.05). The number of alleles ranges from 3 to 11, F_IS_ values from −1.16 to 0.115, Ae from 3.13 to 4.635, the upper limit of the 95% confidence interval of F_IS_ from 0.140 to 0.360, H_e_ from 0.540 to 0.860, H_o_ from 0.535 to 0.635, and the lower limit of the 95% confidence interval of F_IS_ from −0.280 to −0.04. For CBOU, deviations from Hardy–Weinberg equilibrium are observed only within the range of ND and NS. The number of alleles varies from 4 to 11, F_IS_ from −0.35 to 0.17, Ae from 2.73 to 4.53, the upper limit of the 95% confidence interval of F_IS_ from 0.05 to 0.265, H_e_ from 0.09 to 0.805, H_o_ from 0.370 to 0.810, and the lower limit of the 95% confidence interval of F_IS_ from −0.910 to −0.080.

For CC, deviations from Hardy–Weinberg equilibrium cover the range from highly significant (*p* < 0.01) and significant (*p* < 0.05) to non-determined (ND) and non-significant (*p* > 0.05). The number of alleles ranges from 3 to 10.5, F_IS_ values from −0.18 to 0.21, Ae from 1.94 to 4.635, the upper limit of the 95% confidence interval of F_IS_ from −0.02 to 0.36, H_e_ from 0.540 to 0.860, H_o_ from 0.535 to 0.745, and the lower limit of the 95% confidence interval of F_IS_ from −0.91 to −0.57.

For CM, deviations from Hardy–Weinberg equilibrium are observed only within the range of ND and NS (*p* > 0.05). The number of alleles varies from 4 to 11, F_IS_ from −0.15 to 0.135, Ae from 2.63 to 4.53, the upper limit of the 95% confidence interval of F_IS_ from 0.04 to 0.39, H_e_ from 0.29 to 0.86, H_o_ from 0.37 to 0.745, and the lower limit of the 95% confidence interval of F_IS_ from −0.57 to −0.15.

For CR, deviations from Hardy–Weinberg equilibrium span from highly significant (*p* < 0.01) and significant (*p* < 0.05) to non-determined (ND) and non-significant (*p* > 0.05). The number of alleles ranges from 2 to 11, F_IS_ values from −0.635 to 0.17, Ae from 2.71 to 5.64, the upper limit of the 95% confidence interval of F_IS_ from 0.04 to 0.28, H_e_ from 0.09 to 0.86, H_o_ from 0.37 to 0.745, and the lower limit of the 95% confidence interval of F_IS_ from −0.57 to −0.15.

For PI, deviations from Hardy–Weinberg equilibrium cover the range from highly significant (*p* < 0.01) and significant (*p* < 0.05) to non-determined (ND) and non-significant (*p* > 0.05). The number of alleles ranges from 2 to 9.5, F_IS_ values from −0.15 to 0.135, Ae from 2.745 to 5.64, the upper limit of the 95% confidence interval of F_IS_ from 0.07 to 0.28, H_e_ from 0.54 to 0.86, H_o_ from 0.565 to 0.81, and the lower limit of the 95% confidence interval of FIS from −0.57 to −0.08. Differences in deviations from Hardy–Weinberg equilibrium exist among breeds, with CBOU and CM exhibiting non-determined (ND) and non-significant (*p* > 0.05) values, while CB, CC, CR, and PI show values across the range, indicating some alleles deviate from Hardy–Weinberg equilibrium. This phenomenon was observed by Garrido et al. [26] in the Alano Español, detecting a deviation from equilibrium for two microsatellites due to a heterozygous deficit. In their study, 77.5% of alleles were in equilibrium, while 7.2% showed highly significant imbalance, 7.9% significant imbalance, and 7.2% were undetermined.

According to Abramovs et al. [27], Hardy–Weinberg equilibrium is a fundamental principle in population genetics, stating that genotypic frequencies remain constant between generations in the absence of external perturbations. Waples [28] suggests that factors leading to locus-specific deviations from Hardy–Weinberg balance include selective mating, null alleles, genotyping errors, and sexual bonding differences between sexes, leading to excess heterozygosity. He also proposes that identifying problematic loci with deviations could warrant their removal from the dataset for further analysis.

Differences in the number of alleles among breeds are notable, with variations in both their lower and upper limits. The lower limit ranges from 2 (observed in CR and PI) to 4 (seen in CBOU and CM), with CB and CC appearing from 3. Regarding the upper limit, PI peaks at 9.5, CC at 10.5, while the rest—CB, CBOU, CM, and CR—reach 11 alleles. CR exhibits the widest range, from 2 to 11, indicating higher allelic richness, while CM shows the narrowest range, from 4 to 11. Similar findings were observed in the Spanish Alano breed by Garrido et al. [26], with values ranging from 3 to 7, and by San José et al. [29] for the Valencian Hound, with a range of 6–15. Greenbaum et al. [23] assert that allele richness reflects genetic diversity, indicating a population’s potential for adaptability and persistence. Differences in the F_IS_ values among breeds are evident, with variations in both their lower and upper limits. The lowest value is −1.16, observed only in CB, while CR appears from −0.635, CBOU from −0.35, CC from −0.18, and CM and PI from −0.15. On the higher end, CB disappears at 0.115, CM and PI at 0.135, CBOU and CR at 0.17, and CC reaches 0.21. CB exhibits the widest range, at 1.27, higher than values reported for the Bedlington Terrier by Koskinen and Bredbacka [30]. CM and PI have the narrowest range, at 0.285, similar to values for the Wire-Haired Dachshund. All values are higher than those reported for Finnish populations of five dog breeds by Koskinen and Bredbacka [30].

According to Kardos et al. [31], F_IS_ ranges from −1 to 1, with positive values indicating closer average pairings than expected, resulting in a deficit of heterozygotes, and negative values indicating less closely related pairings than expected, leading to an excess of heterozygotes. F_IS_ should not be seen as a measure of individual inbreeding but rather as an indicator of population-level genetic diversity.

Differences in the effective number of alleles (Ae) among breeds are notable, with variations in both their lower and upper limits. The lowest value, 1.94, is observed only in CC, while CM appears from 2.63, CR from 2.71, CBOU from 2.73, PI from 2.745, and CB from 3.13. On the higher end, CBOU and CM disappear at 4.53, CB and CC at 4.635, and CR and PI reach 5.64. CR exhibits the widest range, from 2.71 to 5.64, while CB has the narrowest range, from 3.13 to 4.635. Lower results were reported by Czyż et al. [32] for the Chihuahua, with a range of 3.33 to 5.44, and higher results for the Caucasian Shepherd, with a range of 2.37 to 5.33. According to Greenbaum et al. [23], Ae is a measure of genetic diversity, indicating a population’s potential for adaptability and persistence.

Differences in the upper limit of the 95% confidence interval of the F_IS_ values among breeds are evident, with variations in both their lower and upper limits. The lowest value, −0.02, is observed only in CC, while CM and CR appear from 0.04, CBOU from 0.05, PI from 0.07, and CB from 0.14. On the higher end, CBOU disappears at 0.265, CR and PI at 0.28, CB and CC at 0.36, and CM reaches 0.39. This range of values indicates with 95% probability where the upper limit value of the F_IS_ lies.

Regarding the value of He, differences among breeds exist in both their lower and upper limits. The lowest value, 0.09, is observed in CBOU and CR, while CM appears from 0.29 and CB, CC, and PI from 0.54. On the higher end, CBOU disappears at 0.805, while CB, CC, CM, CR, and PI reach 0.86. CR exhibits the widest range, from 0.09 to 0.86, indicating a higher degree of genetic diversity, while CB has the narrowest range, from 0.54 to 0.86. These findings align with the importance of monitoring genetic diversity for timely identification and improving breeding work on biodiversity in different dog breeds, as highlighted by Dzitsiuk et al. [33]. According to Harris AM [34], expected heterozygosity (He) is a common statistic for assessing genetic variation within populations, with decreased accuracy and precision observed in related or inbred individuals due to greater allele copy dependence in the sample. This range of values aligns with observations by Koskinen and Bredbacka [30] for the Bedlington Terrier and the Wire-Haired Dachshund.

Differences in the observed heterozygosity (Ho) values among breeds are apparent, with variations in both their lower and upper limits. The lowest value, 0.37, is observed in CBOU, CM, and CR, while CB and CC appear from 0.535, and PI from 0.565. On the higher end, CB disappears at 0.635, while CC, CM, and CR vanish at 0.745, and the remaining CBOU and PI breeds reach 0.81. CBOU exhibits the widest range of values, from 0.37 to 0.81, indicating a higher degree of heterozygosity, which contrasts with observations by Koskinen and Bredbacka [30] for the Wire-Haired Dachshund. The narrowest range is observed in CB, from 0.535 to 0.635, a value similar to that reported for the Chihuahua by Czyż et al. [32].

According to Tripp Valdez [35], H_o_ represents the proportion of heterozygous organisms calculated from observed genotypes in a population sample. Animals with high heterozygosity within a breed are less affected by inbreeding, while small population sizes and inbreeding can decrease heterozygosity.

Regarding the value of the lower limit of the confidence interval of the F_IS_, differences among breeds exist in both their lower and upper limits. The lowest value, −0.91, is observed in CBOU and CC, while CM, CR, and PI appear from −0.57, and CB from −0.28. On the higher end, CC disappears at −0.57, while CM and CR vanish at −0.15, and CBOU and PI reach −0.08, with CB reaching −0.04.

## 4. Materials and Methods

### 4.1. Sampling

Sampling was conducted by various officially recognized associations or clubs for the CB, CBOU, PI, CR, CM, and CC breeds (Figure 5A–F). Hair samples were collected from individuals meeting the breed standard or population characteristics (in the case of CC), identified with the animal’s microchip. All Appendix A, including sex, date of birth, genealogies, and owner details, was reported to the DNA Germplasm Bank for Balearic dog breeds in the Applied Molecular Genetics Laboratory (Research Group PAIDI-AGR-218, University of Córdoba). Sampling for each breed occurred over several years to ensure a significant and representative sample size. The number of samples sent per breed was 241 for CB, 72 for CBOU, 94 for PI, 116 for CR, 94 for CM, and 47 for CC, covering all the conducted studies and analyses (genetic characterization, parentage tests, and assignments).

### 4.2. Laboratorial Analyses

A total of 275 samples from the Applied Molecular Genetics Laboratory (Research Group PAIDI-AGR-218, University of Córdoba, Córdoba, Spain) were used for inter-racial studies, encompassing all six breeds. The DNA was extracted using the method described by Walsh et al. [36], utilizing hairs with visible hair bulbs.

The recommended microsatellites by the International Society of Animal Genetics (ISAG) for diversity and parentage studies in dogs were employed. Depending on the breed, 21 or 33 microsatellites were used. Commonly used microsatellites included AHT121, AHT137, AHTh130, AHTh171, AHTh260, AHTk211, AHTk253, CXX279, FH2848, FH2054, INRA021, INU005, INU030, INU055, REN105L03, REN162C04, REN169D01, REN169O18, REN247M23, REN54P11, and REN64E19. Additional microsatellites for CR were 2642RD, 1404RD, 1878RD, 0914RD, 2469RD, 0176RD, 0959RD, 0323RD, 0669RD, 0123RD, 1055RD, and 1257RD.

Microsatellites were amplified using polymerase chain reaction (PCR), and the amplified fragments were separated by capillary electrophoresis using an ABI3130Xl capillary automatic sequencer. Fragment analysis and allelic typing were performed using GENESCAN ANALYSIS v. 3.1.2 and GENOTYPER v. 2.5.2 software, respectively. Size standardization was carried out using Genescan 500 HD LIZ Orange Size standard (Applied Biosystems, Thermo Fisher Scientific, Foster City, CA, USA).

**Figure 5 ijms-25-02706-f005:**
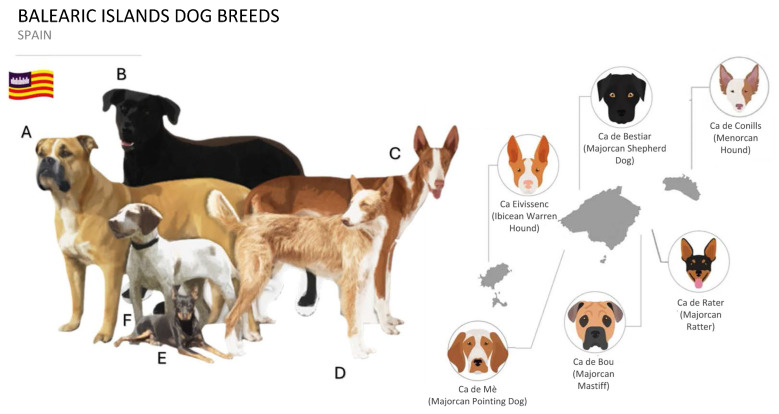
Balearic dog breeds. (**A**) Ca de Bou, CBOU; (**B**) Ca de Bestiar, CB; (**C**) Ca Eivissenc o Podenco Ibicenco, PI; (**D**) Ca de Conills Menorquí, CC; (**E**) Ca de Rater Mallorquí, CR; (**F**) Ca Mè, CM; and province of their major distribution across the Balearic Islands.

### 4.3. Intra-Racial Genetic Diversity

Mean number of alleles per locus (MNA), expected heterozygosity (He), and observed (Ho) heterozygosity were calculated using the Excel MICROSATELLITE TOOLKIT v 3.1. [37]. The intra-racial genetic variation coefficient (F_IS_) with a 95% confidence interval was calculated using GENETIX v. 4.0.5 [38]. Substructure analysis of the studied dog breeds was performed using STRUCTURE v. 2.3.4 [39], dividing individuals into clusters (K) based on similar variation patterns.

For CR and CM breeds, effective number of alleles was calculated using PopGene [40], and polymorphic information content (PIC) was calculated using CERVUS v. 3.0.7.

A Hardy–Weinberg equilibrium test was conducted using GENEPOP v. 1.2.2 [41], applying Fisher’s exact test with the Markov chain Monte Carlo method [42] and Bonferroni correction for CB, CR, CM, and CC breeds.

### 4.4. Inter-Racial Genetic Diversity

All Balearic breeds were included for a comprehensive overview of the genetic situation of each breed relative to others. Wright’s F-statistics [43] were calculated, including FIT (individual inbreeding coefficient relative to the total population), F_ST_ (effect of subpopulations compared to the total population), and F_IS_ (individual inbreeding coefficient relative to its subpopulation) with a 95% confidence interval. These statistics were calculated using GENETIX v. 4.0.5 [38]. A correspondence factor analysis was performed with the same program. Reynolds genetic distances [44] were calculated using POPULATIONS v. 1.2.32 [45]. A distance-based tree was constructed to represent the genetic relationships graphically. The STRUCTURE v. 2.3.4 (Pritchard, Stephens et al. 2000 [39]) was employed, with a Bayesian algorithm, to study the genetic structure of populations, creating as many clusters as existing populations plus one to identify internal structures, if any. For the study of Balearic breeds, this was carried out from K = 2 to K = 7, with 10 repetitions for each K. Finally, the optimal K was calculated according to the method of Evanno et al. [46] using STRUCTURE HARVESTER v. 0.6.94 [47] based on the results obtained from STRUCTURE v. 2.3.4. CLUMPAK v. 1.1 [48] was used for visualizing the results of genetic structure.

### 4.5. Statistical Analyses

#### 4.5.1. Canonical Discriminant Analysis (CDA)

##### CDA Methodology

Canonical discriminant analysis (CDA) was employed as a powerful statistical tool to study the relationships between genetic diversity parameters and the microsatellite molecular markers used in each breed. This involved a comprehensive analysis to create a robust classification tool capable of discerning intricate patterns within and between breeds based on the information available about them. The analysis utilized the Classify package of SPSS version 26.0 software and the canonical discriminant analysis routine of the Analyzing Data package of XLSTAT version 2014.5.03 (Addinsoft Pearson Edition 2014, Addinsoft, Paris, France).

##### Canonical Relationship Plotting

The initial step involved the visualization of canonical relationships to provide a spatial representation of group differences. Leveraging regularized forward stepwise multinomial logistic regression algorithms, variable selection was carried out, considering prior probabilities based on group sizes. This meticulous approach aimed to optimize the efficiency of the subsequent discriminant analyses.

##### Sample Size Consideration

A critical aspect of the study was the adherence to robust sample size practices. Maintaining a ratio of 4–5 times more observations to independent variables ensured statistical power, a crucial element in achieving meaningful and reliable results. This approach followed the recommendations of established research practices, emphasizing the importance of sample size in maintaining analytical rigor.

##### Multicollinearity Analysis

To ensure the integrity of the analyses, an in-depth examination of multicollinearity was conducted. Variance inflation factor (VIF) and tolerance were employed to gauge the linear relationships among predictors. A VIF threshold of 5 was utilized to identify and mitigate potential multicollinearity issues.

##### Canonical Correlation Dimension

The exploration of canonical correlations provided insights into the relationships between sets of variables. Emphasis was placed on canonical correlation values exceeding 0.30, indicating a substantial proportion of explained variance in the dataset.

##### Discriminant Analysis Efficiency

Efficiency in discriminant analysis was gauged through Wilks’ Lambda test, assessing the significance of variables in the discriminant function. The χ^2^ test examined the significance of Wilks’ Lambda, providing crucial insights into the well-explained group adscription.

##### Discriminant Model Reliability

The reliability of the discriminant analysis model was evaluated through Pillai’s trace criterion, specifically suitable for cases of unequal sample sizes. Significance at a level of 0.05 or below indicated the statistical significance of the predictor set in explaining variations in molecular genetic diversity parameters and molecular microsatellite markers across distinct breeds.

##### Canonical Loading Interpretation

Canonical loadings played a pivotal role in the interpretation of discriminating variables. Variables with substantial discriminant loading (≥|0.40|) were identified, contributing significantly to the classification. The stepwise procedure technique ensured the exclusion of non-significant variables.

##### Discriminant Function Reliability: Validation and Crossvalidation

The crossvalidation phase involved leave-one-out analysis, determining the probability of correct breed classification. The evaluation utilized Press’s Q statistic, comparing the classification rate against a critical χ^2^ value. This rigorous assessment provided insights into the generalizability and reliability of the discriminant functions.

#### 4.5.2. Data-Mining CHAID

##### Decision Tree

The CHAID decision tree analysis was conducted using the CHAID package in both SPSS version 26.0 software and XLSTAT version 2014.5.03 (Addinsoft Pearson Edition 2014, Addinsoft, Paris, France).

##### Decision Tree Methodology

The data-mining phase utilized the chi-squared automatic interaction detection (CHAID) decision tree methodology to examine whether the values for genetic diversity parameters and molecular microsatellite markers follow certain patterns across the different breeds in the Balearic Archipelago. CHAID, a technique focused on classification, prediction, and data interpretation, employed a root node, branches, and leaf nodes. Internal nodes were created around variables related to the genetic diversity parameters and molecular microsatellite markers, guided by a chi-square test significance split criterion (*p* < 0.05). Pruning processes, both pre and post, were implemented to prevent overcomplication and ensure the inclusion of branches significantly contributing to the overall fit. The decision tree, analogous to forward stepwise regression, aimed to capture significant relationships among independent variables. Each branch represented outcomes of the test, while leaf nodes indicated category levels of the target variable (breed in our case).

##### Decision Tree Reliability: Validation and Crossvalidation

The crossvalidation of the decision tree was essential to validate its generalizability to the parameters of genetic diversity and microsatellite markers. To achieve this, the complexity parameter and crossvalidated error rate were instrumental in selecting a tree that balances accuracy and simplicity. A leave-one-out crossvalidation approach was employed to mitigate overfitting risks and enhance the predictive accuracy of the model for diverse breeds.

In essence, this integrated approach to canonical discriminant analysis and subsequent data-mining methodologies aimed to provide a nuanced understanding of the complex relationships within the dataset. By combining traditional statistical techniques with advanced data-mining methods, the study sought to create a robust and reliable tool for classifying breeds based on the values reported for their molecular genetic diversity parameters and microsatellite markers.

## 5. Conclusions

Each breed showcases a unique genetic profile that reflects its history, breeding practices, and population dynamics. Understanding these aspects is crucial for informed breeding decisions aimed at preserving genetic diversity, promoting breed health, and ensuring long-term sustainability. Ongoing monitoring and management will be essential to address potential challenges and safeguard each breed’s genetic heritage integrity. A moderate range of allelic diversity and deviations from HW equilibrium were revealed for Ca de Bestiar (CB). This suggests a balance between genetic diversity and potential risks associated with inbreeding, highlighting the importance of careful breeding management to maintain genetic health. Similarly, the Ca de Bou (CBOU) exhibits a moderate range of allelic diversity and deviations from equilibrium, indicating potential genetic stability but also the need for ongoing monitoring to prevent genetic drift or homogeneity. Conversely, the Ca de Conills (CC) displays a wider range of allelic diversity and deviations from equilibrium, suggesting a more diverse genetic background. However, this breed also faces challenges related to potential inbreeding and population structure, emphasizing the importance of genetic management strategies to preserve diversity. The Ca Mè (CM) demonstrates genetic characteristics similar to CBOU, with a moderate range of allelic diversity and deviations from equilibrium. This breed may benefit from targeted breeding programs aimed at maintaining genetic diversity and addressing potential inbreeding. In contrast, the Ca de Rater (CR) presents a wider range of allelic diversity and deviations from equilibrium. This highlights the need for proactive breeding practices to safeguard genetic health and preserve unique breed traits. Lastly, the Ca Eivissenc or Ibicean Hound (PI) breed exhibits similar genetic patterns to CC, with a wide range of allelic diversity and deviations from equilibrium. Like CC, PI may require careful genetic management to mitigate the risks of inbreeding and maintain genetic diversity over time. 

## Figures and Tables

**Figure 1 ijms-25-02706-f001:**
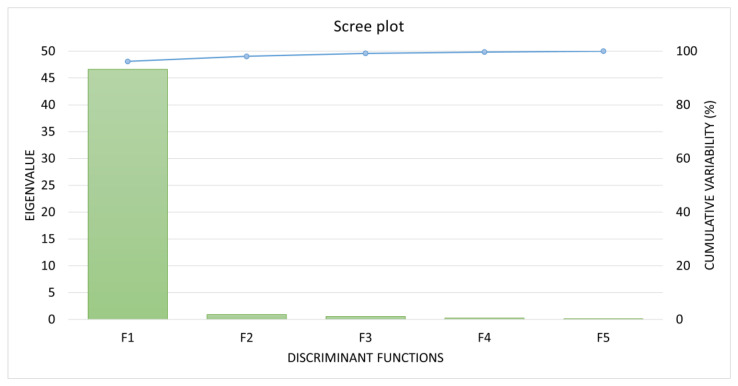
Eigenvalues, discrimination percentages, and cumulative percentages for the revealed set of five discriminant functions (F1 to F5).

**Figure 2 ijms-25-02706-f002:**
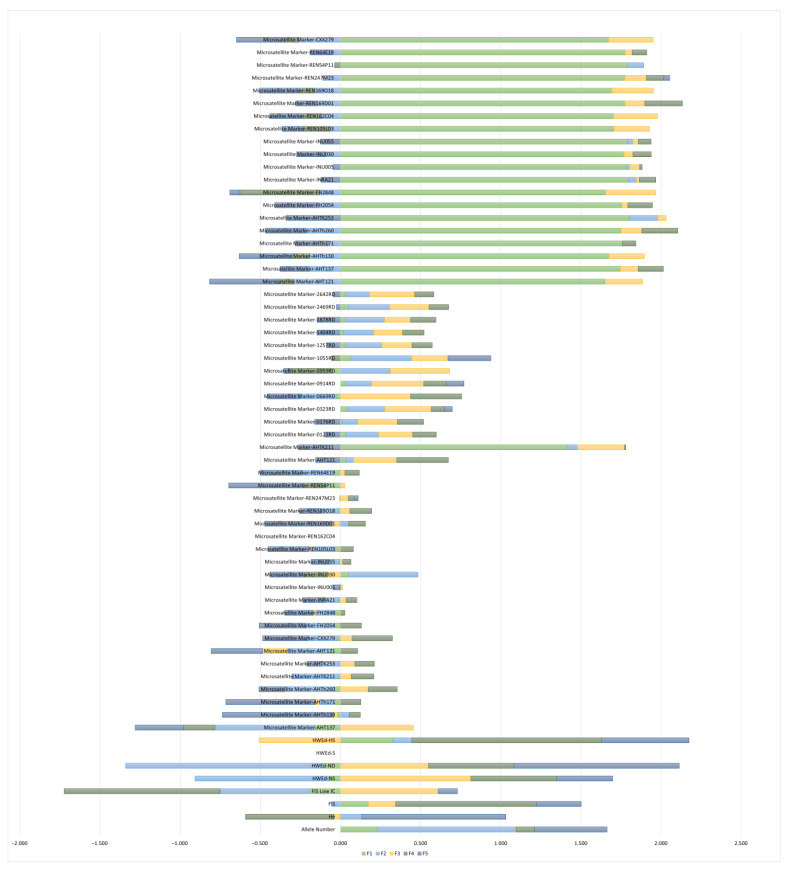
Discriminant loadings for genetic diversity parameters and microsatellite markers across discriminant functions from F1 to F5.

**Figure 3 ijms-25-02706-f003:**
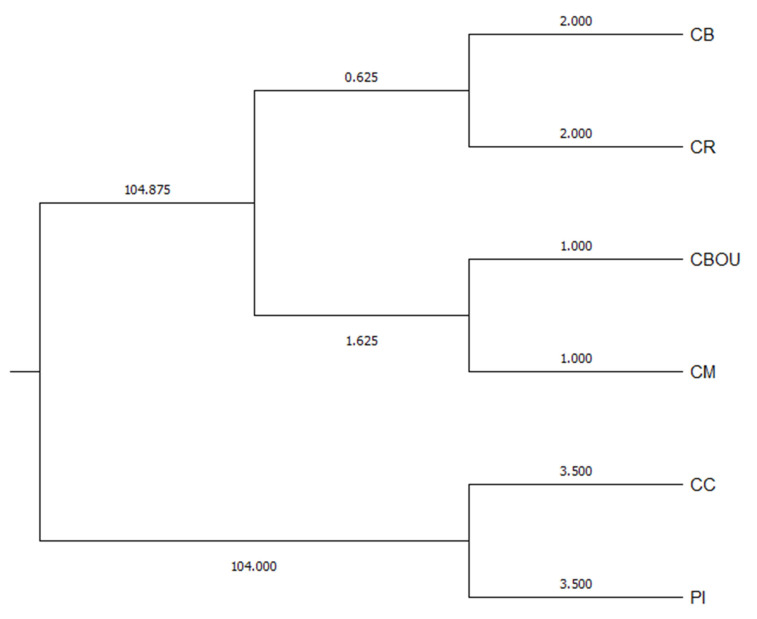
Territorial map representing Mahalanobis distances across endangered dog breeds in the Balearic Islands. Ca de Bestiar (CB), Ca Rater (CR), Ca de Bou (CBOU), Ca Mè (CM), Ca de Conills (CC), and Ibicean Hound or Podenco Ibicenco (PI).

**Figure 4 ijms-25-02706-f004:**
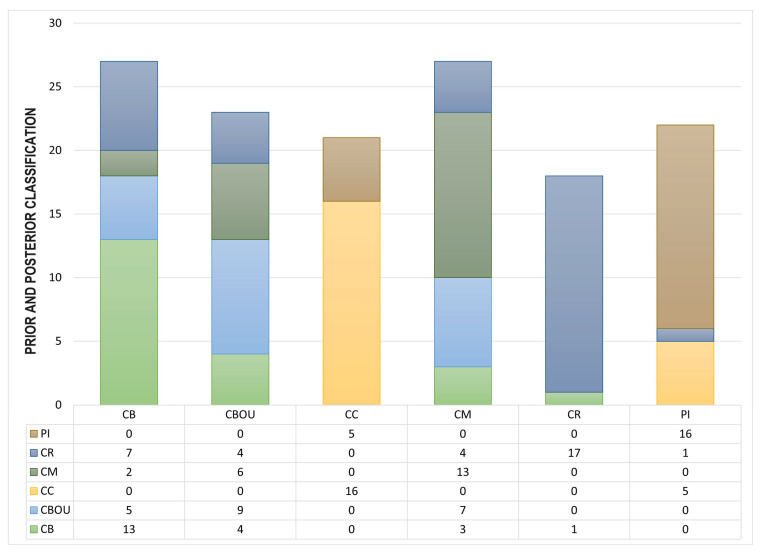
Breeds’ prior and posterior classification, membership probabilities, scores, and squared distances. Ca de Bestiar (CB), Ca Rater (CR), Ca de Bou (CBOU), Ca Mè (CM), Ca de Conills (CC), and Ibicean Hound or Podenco Ibicenco (PI).

## Data Availability

Data is contained within the article and Appendix A.

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
