# Peer review of "Integrated Discriminant Evaluation of Molecular Genetic Markers and Genetic Diversity Parameters of Endangered Balearic Dog Breeds"

_ijms, 2024, doi:10.3390/ijms25052706_

Round 1
Reviewer 1 Report
Comments and Suggestions for Authors
The research is devoted to the important problem of the biodiversity conservation of domestic animals, in particular, unique local breeds of dogs. To develop conservation programs, it is necessary to understand the genetic structure of existing populations, the degree of inbreeding in them, and the identification of specific genetic markers. The authors have carried out a large amount of genetic research on endangered Balearic dog breeds using modern methods. The results are presented and analyzed in detail. As a note:
1. The Abstract must reflect the content of the article, including the results of the study.This section should be supplemented with the obtained digital data. This edition does not show the large volume of interesting research carried out.
2. In Discussion section, your should discuss your own experimental data in comparison with the data of other researches. In this article, the Discussion includes a large amount of information (lines 419-496), which, in my opinion, should be moved to the Introduction section.
3. The Conclusion section in this article contains mainly general considerations, but 6 dog breeds were studied, each of which has its own specific characteristics. The conclusions should be specified so that the research results can be used by breeders working with a particular breed.
4. In Figure 4, the Y axis should be labeled in English.
Author Response
Reviewer: 1
The research is devoted to the important problem of the biodiversity conservation of domestic animals, in particular, unique local breeds of dogs. To develop conservation programs, it is necessary to understand the genetic structure of existing populations, the degree of inbreeding in them, and the identification of specific genetic markers. The authors have carried out a large amount of genetic research on endangered Balearic dog breeds using modern methods. The results are presented and analyzed in detail.
Response: We thank the reviewer for his/her kind comments.
As a note:
- The Abstract must reflect the content of the article, including the results of the study.This section should be supplemented with the obtained digital data. This edition does not show the large volume of interesting research carried out.
Response: We followed the reviewer suggestion.
- In Discussion section, your should discuss your own experimental data in comparison with the data of other researches. In this article, the Discussion includes a large amount of information (lines 419-496), which, in my opinion, should be moved to the Introduction section.
Response: We understand the reviewer comment as in the manner the paragraph was introducted, it seemed to be aprt of the introduction. However, these are the results and explanations for multicollinearity analyses. We clarified this in the body text in order to make it fit the style in which discussion should eb written.
- The Conclusion section in this article contains mainly general considerations, but 6 dog breeds were studied, each of which has its own specific characteristics. The conclusions should be specified so that the research results can be used by breeders working with a particular breed.
Response: Reviewer suggestion was followed.
- In Figure 4, the Y axis should be labeled in English.
Response: Reviewer suggestion was followed.
Reviewer 2 Report
Comments and Suggestions for Authors
Manuscript entitled “Integrated Discriminant Evaluation of Molecular Genetic Markers and Genetic Diversity Parameters of Endangered Balearic Dog Breeds” studies six endangered dog breeds present in the Balearic Islands, through the use of microsatellite markers
The manuscript is well structured and well written and deserves to be published on IJMS.
The text is very extensive, especially the discussion section; it could be reduced, as well as the results section could be reduced with the use of tables.
The authors should state in their manuscript that microsatellite analysis is not the most up-to-date method for analyzing and comparing the dog genome.
Minor suggestions
please add some pictures of the dog breeds analyzed
please indicate the version, each time you use a software tool
please write the dog species in italics
Author Response
Reviewer 2.
Manuscript entitled “Integrated Discriminant Evaluation of Molecular Genetic Markers and Genetic Diversity Parameters of Endangered Balearic Dog Breeds” studies six endangered dog breeds present in the Balearic Islands, through the use of microsatellite markers
The manuscript is well structured and well written and deserves to be published on IJMS.
Response: We thank the reviewer for his/her kind comments.
The text is very extensive, especially the discussion section; it could be reduced, as well as the results section could be reduced with the use of tables.
Response: We reduced the discussion section 4304 words to 2691 words.
The authors should state in their manuscript that microsatellite analysis is not the most up-to-date method for analyzing and comparing the dog genome.
Response: We followed reviewer suggestion.
Minor suggestions
please add some pictures of the dog breeds analyzed
Response: Figure 5 was added.
please indicate the version, each time you use a software tool
Response: We followed the reviewer suggestion.
please write the dog species in italics
Response: We followed the reviewer suggestion.